

# A scalable discrete-time survival model for neural networks

Michael F. Gensheimer[1] and Balasubramanian Narasimhan[2]

[1] Department of Radiation Oncology, Stanford University, Stanford, CA, United States of America
[2] Department of Statistics, Stanford University, Stanford, CA, United States of America

## ABSTRACT

There is currently great interest in applying neural networks to prediction tasks in medicine. It is important for predictive models to be able to use survival data, where each patient has a known follow-up time and event/censoring indicator. This avoids information loss when training the model and enables generation of predicted survival curves. In this paper, we describe a discrete-time survival model that is designed to be used with neural networks, which we refer to as Nnet-survival. The model is trained with the maximum likelihood method using mini-batch stochastic gradient descent (SGD). The use of SGD enables rapid convergence and application to large datasets that do not fit in memory. The model is flexible, so that the baseline hazard rate and the effect of the input data on hazard probability can vary with follow-up time. It has been implemented in the Keras deep learning framework, and source code for the model and several examples is available online. We demonstrate the performance of the model on both simulated and real data and compare it to existing models Cox-nnet and Deepsurv.

# INTRODUCTION

With the popularization of deep learning and the increasing size of medical datasets, there has been increasing interest in the use of machine learning to improve medical care. Several recent papers have described use of neural network or other machine learning techniques to predict future clinical outcomes (*Rajkomar et al., 2018*; *Kwong et al., 2017*; *Miotto et al., 2016*; *Avati et al., 2017*). The outcome measure is generally evaluated at one follow-up time point, and there is often little discussion of how to deal with censored data (e.g., patients lost to follow-up before the follow-up time point). This is not ideal, as information about censored patients is lost and the model would need to be re-trained to make predictions at different time points. Because of these issues, modern predictive models generally use Cox proportional hazards regression or a parametric survival model instead of simpler methods such as logistic regression that discard time-to-event information (*Cooney, Dudina & Graham, 2009*).

Several authors have described solutions for modeling time-to-event data with neural networks. These are generally adaptations of linear models such as the Cox proportional hazards model (*Cox, 1972*). Approaches include a discrete-time survival model with

Corresponding author
Michael F. Gensheimer,
mgens@stanford.edu

a heuristic loss function (*Brown, Branford & Moran, 1997*), a parametric model with predicted survival time having a Weibull distribution (*Martinsson, 2016*), and adaptations of the Cox proportional hazards model (*Faraggi & Simon, 1995*; *Ching, Zhu & Garmire, 2018*; *Katzman et al., 2018*). Most of the models assume proportional hazards (the effect of each predictor variable is the same at all values of follow-up time). This is not a very realistic assumption for most clinical situations. In the past, when models were typically trained using dozens or hundreds of patients, it was often not possible to demonstrate violation of proportional hazards. However, in the modern era of datasets of thousands or millions of patients, it will usually be possible to demonstrate violation of the proportional hazards assumption, either by plotting residuals or with a statistical test.

In this paper, we describe Nnet-survival, a discrete-time survival model that is theoretically justified, naturally deals with non-proportional hazards, and is trained rapidly by mini-batch gradient descent. It may be useful in several situations, especially when non-proportional hazards are known to be present, for very large datasets that do not fit in memory, or when predictor data is a good fit for a neural network approach (such as image or text data). We have published source code for the use of the model with the Keras deep learning library, which is available at http://github.com/MGensheimer/nnet-survival.

## MATERIALS & METHODS

### Relationship to prior work

In this section we describe prior approaches to the problem and illustrate some pitfalls that are addressed with our model.

Several authors have adapted the Cox proportional hazards model to neural networks (*Faraggi & Simon, 1995*; *Ching, Zhu & Garmire, 2018*; *Katzman et al., 2018*). This is potentially attractive since the Cox model has been shown to be very useful and is familiar to most medical researchers. One issue with this approach is that the partial likelihood for each individual depends not only on the model output for that individual, but also on the output for all individuals with longer survival. This would preclude the use of stochastic gradient descent (SGD) since with SGD only a small number of individuals are visible to the model at a time. Therefore, the entire dataset would need to be used for each gradient descent step. This is undesirable because it slows down convergence, cannot be applied to datasets that do not fit into memory ("out-of-core learning"), and could result in getting stuck in a local minimum of the loss function (*Bottou, 1991*).

An alternative approach that avoids the above issue is to use a fully parametric survival model, such as a discrete time model. See Section 7.5 of *Rodriguez (2016)* for a brief overview of discrete time survival models. Brown et al. proposed a discrete-time survival model using neural networks (*Brown, Branford & Moran, 1997*). This model can easily be trained with SGD, which is attractive. Follow-up time is divided into a set of fixed intervals. For each time interval the conditional hazard probability is estimated: the probability of failure in the interval, given that the individual has survived at least to the beginning of the interval. For each time interval $j$, the neural network loss is defined as (adapted from Eq.

17 in *Brown, Branford & Moran 1997)*:

$$\frac{1}{2}\sum_{i=1}^{d_j}(1-h_j^i)^2 + \frac{1}{2}\sum_{i=d_j+1}^{r_j}(h_j^i)^2 \qquad (1)$$

where $h_j^i$ is the hazard probability for individual $i$ during time interval $j$, there are $r_j$ individuals "in view" during the interval $j$ (i.e., have not experienced failure or censoring before the beginning of the interval) and the first $d_j$ of them suffer a failure during this interval. The overall loss function is the sum of the losses for each time interval.

The authors note that in the case of a null model with no predictor variables, minimizing the loss in Eq. (1) results in an estimate of the hazard probabilities that equals the Kaplan–Meier maximum likelihood estimate: $\hat{h}_j = \frac{d_j}{r_j}$. While this is true, the equivalence does not hold once each individual's hazard depends on the value of predictor variables.

A more theoretically justified loss function, which we use in our model, would be the negative of the log likelihood function of a statistical survival model. This likelihood function has been well studied for discrete-time survival models in a non-deep learning context. Adapting Eq. (3.4) from *Cox & Oakes (1984)* and Eq. (2.17) from *Singer & Willett (1993)*, the contribution of time interval $j$ to the overall log likelihood is:

$$\sum_{i=1}^{d_j}\ln(h_j^i) + \sum_{i=d_j+1}^{r_j}\ln(1-h_j^i). \qquad (2)$$

This is similar but not identical to Eq. (1) and can be shown to produce different values of the model parameters for anything more complex than the null model (for an example, see the file `brown1997_loss_function_example.md` in our GitHub repository).

The proposed model using Eq. (2) naturally incorporates time-varying baseline hazard rate and non-proportional hazards if each time interval output node is fully connected to the last hidden layer's neurons. The neural network has $n$-dimensional output where $n$ is the number of time intervals, giving a separate hazard rate for each time interval.

There are several attractive features of the proposed model:
1. It is theoretically justified and fits into the established literature on survival modeling
2. The loss function depends only on the information contained in the current mini-batch, which enables rapid training with mini-batch SGD and application to arbitrary-size datasets
3. It is flexible and can be adapted to specific situations. For instance, for small sample size where we wish to minimize the number of neural network parameters, it is easy to incorporate a proportional hazards constraint so that the effect of the input data on the hazard function does not vary with follow-up time.

## Model formulation

Follow-up time is divided into $n$ intervals which are left-closed and right-open. Let $[t_1, t_2, \ldots, t_n]$ be the times at the upper limit of each interval. The conditional hazard probability $h_j$ is defined as the probability of failure in interval $j$, given that the individual has survived at least to the beginning of the interval. $h_j$ can vary per individual according to

the input and the weights of the neural network. The predicted probability of an individual surviving at least to the end of interval $j$ is:

$$S_j = \prod_{i=1}^{j}(1 - h_i). \tag{3}$$

The model likelihood can be divided either by time interval as in Eq. (2), or by individual. For a neural network trained with mini-batches of individuals, the latter formulation translates more easily into computer code. For an individual with failure during interval $j$ (i.e., uncensored), the likelihood is the probability of surviving through intervals 1 through $j-1$, multiplied by the probability of failing during interval $j$:

$$lik = h_j \prod_{i=1}^{j-1}(1 - h_i) \tag{4}$$

$$loglik = ln(h_j) + \sum_{i=1}^{j-1}\ln(1 - h_i). \tag{5}$$

For a censored individual with a censoring time $t_c$ which falls in the second half of interval $j-1$ or the first half of interval $j$ (i.e., $\frac{1}{2}(t_{j-2} + t_{j-1}) \le t_c < \frac{1}{2}(t_{j-1} + t_j)$), the likelihood is the probability of surviving through intervals 1 through $j-1$:

$$lik = \prod_{i=1}^{j-1}(1 - h_i) \tag{6}$$

$$loglik = \sum_{i=1}^{j-1}\ln(1 - h_i). \tag{7}$$

It can be seen that individuals with a censoring time in the second half of an interval are given "credit" for surviving that interval (without this, there would be a downward bias on the survival estimates (*Brown, Branford & Moran, 1997*).

The full log likelihood of the observed data is the sum of the log likelihoods for each individual. In the neural network survival model, we wish to maximize the likelihood, so we set the loss to equal the negative log likelihood and minimize the loss by by stochastic gradient descent or mini-batch gradient descent.

## Determination of hazard probability

For each time interval, the hazard probability will vary according to the input data. We have implemented two approaches to mapping input data to hazard probabilities:

With the **flexible** version, the final hidden layer (e.g., the "Max pooling" layer in Fig. 1) is densely connected to the output layer (the "Fully connected" layer in Fig. 1). The output layer has $n$ neurons, where $n$ is the number of time intervals. The log odds of surviving each time interval is equal to the dot product of the incoming values and the kernel weights, plus the bias weight. Then, using a sigmoid activation function, log odds are converted to the conditional probability of surviving this interval. With this approach, both the baseline

hazard rate and the effect of input data on the hazard rate can vary freely with follow-up time. This approach is most appropriate for larger datasets or when the proportional hazards assumption is known to be violated.

With the **proportional hazards** version, the baseline hazard probability is allowed to vary freely with time interval, but the effect of input data on hazard rate does not vary with follow-up time (if a certain combination of input data results in a high rate of death in the early follow-up period, it will also result in a high rate of death in the late follow-up period). This is implemented by setting the final hidden layer to have a single neuron, and densely connecting the prior hidden layer to the final hidden layer without any bias weights. The final hidden layer neuron value is $X\beta$, where $X$ is the value of the prior hidden layer neurons and $\beta$ is the weights between the prior hidden layer and the final hidden layer. The $X\beta$ notation is meant to echo that of the "linear predictor" in standard survival analysis, for instance section 18.2 of *Harrell Jr (2015)*. The conditional probability of surviving the interval $i$ is (adapted from Eq. (18.13) in *Harrell Jr (2015)*:

$$1 - h_i = (1 - h_{base})^{exp(X\beta)} \tag{8}$$

where $h_{base}$ is the baseline hazard probability for this time interval. The $h_{base}$ values are estimated as part of the neural network by training a set of $n$ weights, which are each transformed by a sigmoid function to convert baseline log odds of surviving each time interval into baseline probability of survival. These sigmoid-transformed weights, along with the final hidden layer value, contribute to the $n$-dimensional output layer according to Eq. (8). See class *PropHazards* in file `nnet_survival.py` in the GitHub repository. The proportional hazards approach is useful for small datasets where one wishes to reduce overfitting by minimizing the number of parameters to optimize. It also makes it easier to interpret the reasons for the model's predictions. This version is very similar to a traditional proportional hazards discrete-time survival model using a complementary log–log link (see *Rodriguez (2016)*, section 7.5.3: "Discrete Survival and the C-Log-Log Link").

## Implementation

We implemented Nnet-survival in the Python language, using the Keras library with Tensorflow backend (code at http://github.com/MGensheimer/nnet-survival). A custom loss function is used which represents the negative log likelihood of the survival model. The output of the neural network is an $n$-dimensional vector $surv_{pred}$, where $n$ is the number of time intervals. Each element represents the predicted conditional probability of surviving that time interval, or $1 - h_j$. An individual's predicted probability of surviving through the end of time interval $j$ is given by Eq. (3). An example neural network architecture using the "flexible" version of the discrete time survival model is shown in Fig. 1.

Each individual used to train the model has a known failure/censoring time $t$ and censoring indicator, which are transformed into a vector format for use in the model. Vector $surv_s$ has length $n$ and represents the time intervals the individual has survived through; vector $surv_f$ also has length $n$ and represents the time interval during which failure occurred, if it occurred.
**A**

**Input:** 128x1 feature vector
**Output:** Survival probability with time intervals (in days):
[0,10), [10,20), [20,30), [30,40), [40,50)

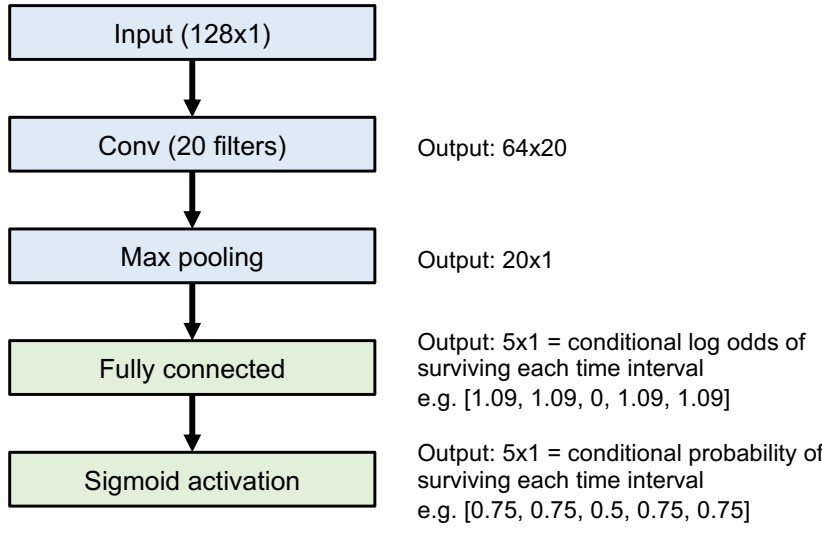

| Input (128x1) |
| --- |

Conv (20 filters) — Output: 64x20

Max pooling — Output: 20x1

Fully connected — Output: 5x1 = conditional log odds of surviving each time interval e.g. [1.09, 1.09, 0, 1.09, 1.09]

Sigmoid activation — Output: 5x1 = conditional probability of surviving each time interval e.g. [0.75, 0.75, 0.5, 0.75, 0.75]

**B**

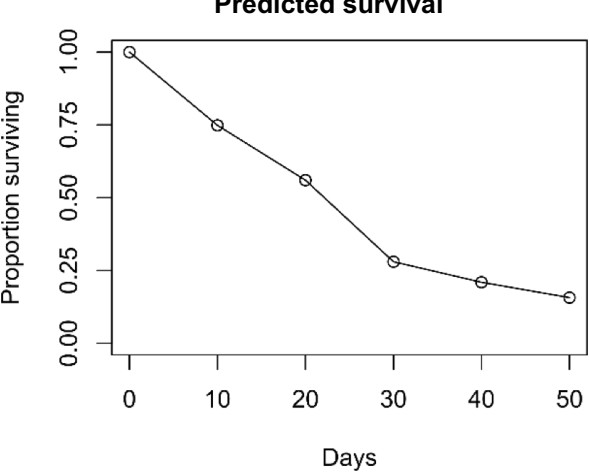

**Figure 1** **Example neural network architecture (A) and output for one individual (B).** Layers in blue are unique to the example neural network; layers in green are common to all neural networks that use the "flexible" version of our survival model.

For individuals with failure (uncensored), for time interval $j$:

$$surv_s(j) = \begin{cases} 1, & \text{if } t \geq t_j \\ 0, & \text{otherwise} \end{cases} \tag{9}$$

$$surv_f(j) = \begin{cases} 1, & \text{if } t_{j-1} \leq t < t_j \\ 0, & \text{otherwise} \end{cases} \tag{10}$$

For censored individuals:

$$surv_s(j) = \begin{cases} 1, & \text{if } t \geq \frac{1}{2}(t_{j-1} + t_j) \\ 0, & \text{otherwise} \end{cases} \tag{11}$$

and

$$surv_f(j) = 0. \tag{12}$$

The log likelihood for each individual is:

$$loglik = \sum_{i=1}^{n} \begin{pmatrix} \ln\left(1 + surv_s(i) \cdot (surv_{pred}(i) - 1)\right) \\ + \ln\left(1 - surv_f(i) \cdot surv_{pred}(i)\right) \end{pmatrix} \tag{13}$$

which is a restatement of Eqs. (5) and (7) to work with the vector encoding of actual and predicted survival.

The loss function is the negative of Eq. (13). The loss function is minimized using gradient descent; Keras performs automatic differentiation of the loss function in order to calculate the gradient. In our experiments, using the custom loss function extended running time very slightly compared to standard loss functions such as mean squared error.

The cut-points for the time intervals can be varied according to the specific application. In most of our experiments we have used 15–40 time intervals, spaced out more widely with increasing follow-up time. This ensures that around the same number of survival events fall into each time interval, which may help ensure reliable estimates for all time intervals. Other authors have suggested using at least ten time intervals to avoid bias in the survival estimates (*Breslow & Crowley, 1974*). In our experiments we have found that the model's performance is fairly robust to choice of specific cut-points.

## Performance evaluation: simulated data

We ran several experiments with simulated data to assess correctness of the model. The code is available in `nnet_survival_examples.py` in the GitHub repository.

### *Simple model with one predictor variable*

We first tested a very simple survival model with one binary predictor variable. Five thousand simulated patients were created. Half of the patients had predictor variable value of 0 and were the poor prognosis patients. For this group, survival times were drawn from an exponential distribution with median survival of 200 days. The other half of the patients had predictor variable value of 1 and were the good prognosis patients. Their survival

times were drawn from an exponential distribution with median survival of 400 days. For both groups, some patients were censored; censoring time was drawn from an exponential distribution with median value / half-life of 400 days. This survival model used the flexible version of nnet-survival (i.e., non-proportional hazards) with no hidden layers and 39 time intervals spanning the range of 0 to 1,780 days.

To evaluate the correctness of this model, we created calibration curves: we plotted and compared actual vs. model-predicted survival curves for the two groups. For each of the two groups, a Kaplan–Meier survival curve was plotted to show actual survival. Then, for each group, a model-predicted survival curve was generated: for each follow-up time point, the average of predicted survival for all patients in that group was calculated and displayed.

### Optimal width of time intervals

We investigated whether model performance depended on time interval width. Similarly to the prior example, we simulated a population of 5,000 patients with one binary predictor variable. Survival time distribution was generated using a Weibull distribution, with scale parameter depending on the predictor variable value. Median survival time for the overall population was 182 days. We used the flexible version of nnet-survival to predict survival time. Four options for time intervals were evaluated:

- Uniform intervals with width of 1 year
- Uniform intervals with width of 1 month
- Uniform intervals with width of 1 week
- Increasing width of intervals with increasing follow-up time, with half-life for interval width of 1 year. Specifically, the time interval borders were placed at: $\frac{-ln(1-x)\cdot 365}{ln(2)}$ for $x$ in $[0.0, 0.05, 0.10, \ldots, 0.95]$

Discrimination performance was assessed with Harrell's C-index.

### Convolutional neural network for MNIST dataset

One area in which neural networks have shown clear superiority to other model types is in analysis of 2D image data, for which convolutional neural networks provide state-of-the-art results. We wished to demonstrate use of a convolutional neural network as part of a survival model. For this, we used the MNIST dataset (*Lecun et al., 1998*). This dataset includes images of 70,000 handwritten digits with gold-standard labels, divided into a training set of 60,000 digits and a test set of 10,000 digits. We created a simulated scenario in which each image corresponds to one patient, and patients with higher digits tend to have shorter survival. The images could be imagined as an X-ray images of tumors, with higher digits representing larger, more deadly tumors. The goal of the model is to predict survival distribution for each test set patient.

We used only images with digits 0 through 4, leaving 30,596 training set images and 5,139 test set images. Image size was $28 \times 28$ pixels. Patients' survival times were drawn from an exponential distribution with scale parameter depending on digit. The scale parameter (with units of days) was set to

$$\beta = \frac{365 \cdot exp(-0.9 \cdot digit)}{ln(2)} \tag{14}$$
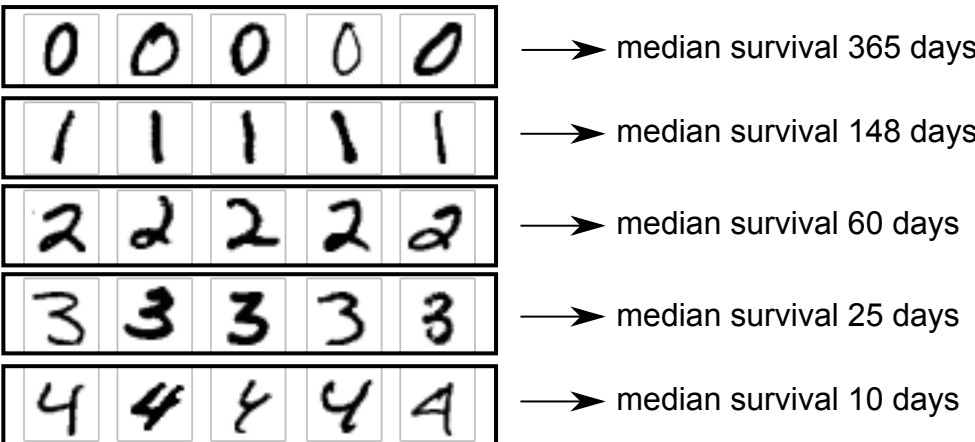

**Figure 2 MNIST dataset construction.** Images of handwritten digits 0–4 were used as predictor of survival time (one image per patient). Actual survival time was generated from an exponential distribution with scale depending on the digit. Lower digits have longer median survival.

with the probability density function being:

$$f(t; \frac{1}{\beta}) = \frac{1}{\beta} \exp(-\frac{t}{\beta}). \tag{15}$$

Therefore, median survival ranged from 365 days for digit 0 down to 10 days for digit 4. The setup is illustrated in Fig. 2.

A five-layer neural network architecture was used, with two convolutional layers of kernel size $3 \times 3$, followed by a max-pooling layer, a fully connected layer of size 4 neurons, then the output layer. The flexible version of the nnet-survival model was used, so that non-proportional hazards were possible. The Adam optimizer was used. Model performance was evaluated using the C-index to measure discrimination, and calibration curves (actual vs. predicted survival curves) to evaluate calibration. As the Nnet-survival model is flexible and the predicted survival curve for each patient can have a different shape, there is no unique ordering of patients by prognosis (i.e., when comparing two patients, one could have a higher probability of 1-year survival but the other could have a higher probability of 2-year survival). Therefore, to calculate C-index, the model's predicted probability of 1-year survival was used to rank the patients.

## Performance evaluation: SUPPORT study (real data)

We evaluated the performance of the Nnet-survival model and other similar models using real patient data. We wished to use a publicly available dataset with time-to-event data on a large number of patients. With a large sample size, we could use data splitting to formally test model performance, and would also be able to evaluate for violations of the proportional hazards assumption of the standard Cox proportional hazards model.

For the real dataset, we chose to use the Study to Understand Prognoses and Preferences for Outcomes and Risks of Treatments (SUPPORT) (*Knaus et al., 1995*). In this multicenter study, 9,105 hospitalized patients had detailed data recorded including diagnoses,

laboratory values, follow-up time and vital status. The dataset is publicly available on the Vanderbilt Biostatistics web site. The task for the survival models was to predict each patient's life expectancy with good discrimination and calibration.

Some patients had missing values for one or more predictor variables; in this case we imputed the missing data by using the median value in the sample, or for laboratory values, using the recommended default value listed on the Vanderbilt Biostatistics web site. If more than 4,000 patients were missing a value for the variable, that variable was excluded from analysis. After processing, there were 39 predictor variables. Patients were divided with a 70%/30% split into training and test sets. The processed dataset is available at our project's GitHub page.

We tested four models on the SUPPORT study dataset:

- Our model, Nnet-survival (flexible version, so that non-proportional hazards were possible)
- Cox-nnet (*Ching, Zhu & Garmire, 2018*)
- Deepsurv (*Katzman et al., 2018*)
- Standard Cox proportional hazards model

All three neural network models used a simple multilayer perceptron architecture with a single hidden layer. The Cox-nnet default parameters specify a hidden layer size of 7 neurons when input dimension is 39, which we felt to be a reasonable choice, so a hidden layer size of 7 was used for the three models. For all three neural network models, L2 regularization was used to help prevent overfitting. The regularization strength parameter was chosen using 10-fold cross validation on the training set, using log likelihood as the performance metric. No regularization was used for the standard Cox proportional hazards model. For Nnet-survival, 19 follow-up time intervals were used, extending out to 6 years (around the maximum follow-up time of the SUPPORT study), with larger spacing for later intervals due to the decreased density of failure events with increasing follow-up time. The RMSprop optimizer was used for Nnet-survival.

As Cox-nnet and Deepsurv only output a prognostic index for each patient, not a predicted survival curve, we generated predicted survival curves for these methods by using the Breslow method to generate a baseline hazard function (*Breslow, 1974*).

To evaluate the models' discrimination performance, we used Harrell's C-index to assess discrimination. To calculate C-index for Nnet-survival, the model's predicted probability of 1-year survival was used to rank the patients.

To evaluate model calibration, we used a published adaptation of the Brier score for censored data (*Graf et al., 1999*). This was implemented using the *ipred* R package. We also created calibration plots for specific follow-up times (*Royston & Altman, 2013*).

We tested the running time of each method by fitting each model using a range of dataset sizes. Simulated datasets ranging from 1,000 to 1,000,000 patients were created by sampling from the 9,105 SUPPORT study patients with replacement. Each combination of model and sample size was run three times and the results were averaged. Each model was trained for 1,000 epochs. An Ubuntu Linux server with 3.6 GHz Intel Xeon E5-1650

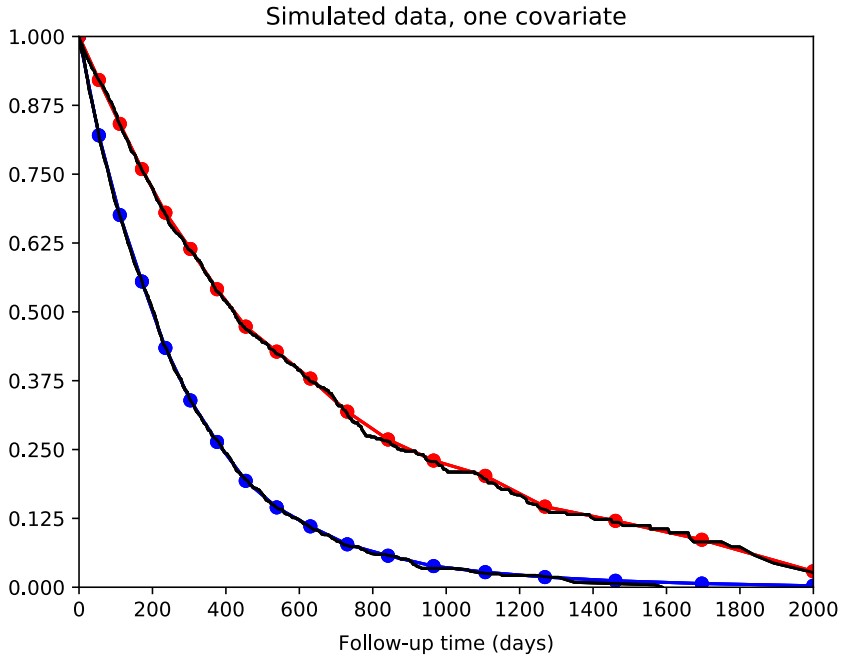

**Figure 3 Simple survival model with one predictor variable.** 5,000 simulated patients with exponential survival distribution. Half of patients have predictor variable value of 0 with median survival of 200 days; the other half have value of 1 with median survival of 400 days. Actual survival for the two groups is shown in black (Kaplan–Meier curves). The average model predictions for the two groups are shown in blue and red, respectively. Model predictions correspond well to actual survival.

CPUs and 32GB of RAM was used. The models were constrained to run on one CPU core. Python version 3.5.2 was used for the Nnet-survival, Cox-nnet, and standard Cox proportional hazards models; Python version 2.7.12 was used for Deepsurv. R version 3.4.3 was used to calculate Brier scores (*R Core Team, 2017*). The code for the SUPPORT study analysis is available in `support_study.py` in the GitHub repository.

# RESULTS

## Simulated data

### Simple model with one predictor variable

We tested a simple survival model with one binary predictor variable and no hidden layers. The calibration curves for the two groups of patients are shown in Fig. 3. It can be seen that calibration is excellent: the actual and predicted survival curves are superimposed.

Model convergence was found to be reliable. The model was optimized repeatedly with different random starting weights and converged to very similar final loss/likelihood values.

### Optimal width of time intervals

The discrimination performance of the survival model was robust to various choices of time interval width and configuration (constant width, or increasing width with increasing follow-up time). For each of the four time interval options, discrimination performance was identical with C-index of 0.66.

**Table 1  Performance of four models on SUPPORT study test set ($n = 2,732$).** Discrimination assessed with C-index. Calibration assessed with modified Brier score (*Graf et al., 1999*) at three specific follow-up times.

| Model | C-index | Brier score: 6 months | Brier score: 1 year | Brier score: 3 years |
|---|---|---|---|---|
| Nnet-survival | 0.732 | 0.181 | 0.184 | 0.177 |
| Cox-nnet | 0.735 | 0.183 | 0.185 | 0.177 |
| Deepsurv | 0.730 | 0.184 | 0.187 | 0.179 |
| Cox PH | 0.734 | 0.183 | 0.186 | 0.178 |

### Convolutional neural network for MNIST dataset

We used the MNIST dataset of handwritten digits to simulate a scenario where each patient has an X-ray image of a tumor, and survival time distribution depends on the appearance of the tumor. Digits 0 through 4 were used, with lower digits having longer median survival. The model's task was to accurately predict survival time for each patient. There were 30,596 images in the training set used to train the model's weights, and 5,139 in the test set used to evaluate model performance.

The Nnet-survival model produced good performance. C-index for the test set was 0.713, compared to 0.770 for a "perfect" model that used the true digit as the predictor variable. Calibration was excellent, as seen in Fig. 4.

### Support study (real data)

Four survival models (Nnet-survival, Cox-nnet, Deepsurv, and a standard Cox proportional hazards model) were tested using the SUPPORT study dataset of 9,105 hospitalized patients.

We found that several predictor variables violated the proportional hazards assumption of the standard Cox model, with an example given in Fig. 5. This provides an opportunity for our Nnet-survival model to have improved calibration compared to the other three models.

All models were trained/fit using the 70% of patients in the training set ($n = 6,373$). Then, performance was measured using the remaining 30% of patients in the test set ($n = 2,732$). Discrimination performance was very similar for all models, with test set C-index around 0.73 (Table 1). Table 1 also shows calibration performance as measured by the Brier score (lower is better). Nnet-survival had the best calibration performance at all three follow-up time points, though the differences were fairly small. Calibration was also assessed visually using calibration plots (Fig. 6). Our Nnet-survival model appeared to have the best calibration at the 6 month and 1 year time points, with Cox-nnet and the standard Cox model tending to under-predict survival probability for the best-prognosis patients.

We compared running time of the three neural network models for various training set sizes, with results shown in Fig. 7. Simulated datasets of size 1,000 to 1,000,000 were created by sampling from the SUPPORT study dataset with replacement. Each model was run for 1,000 epochs. For sample sizes of 100,000 and higher, the Cox-nnet model ran out of memory on a computer with 32 GB memory; therefore, for this model running times could only be calculated for sample sizes of 1,000 to 31,622.

**A**

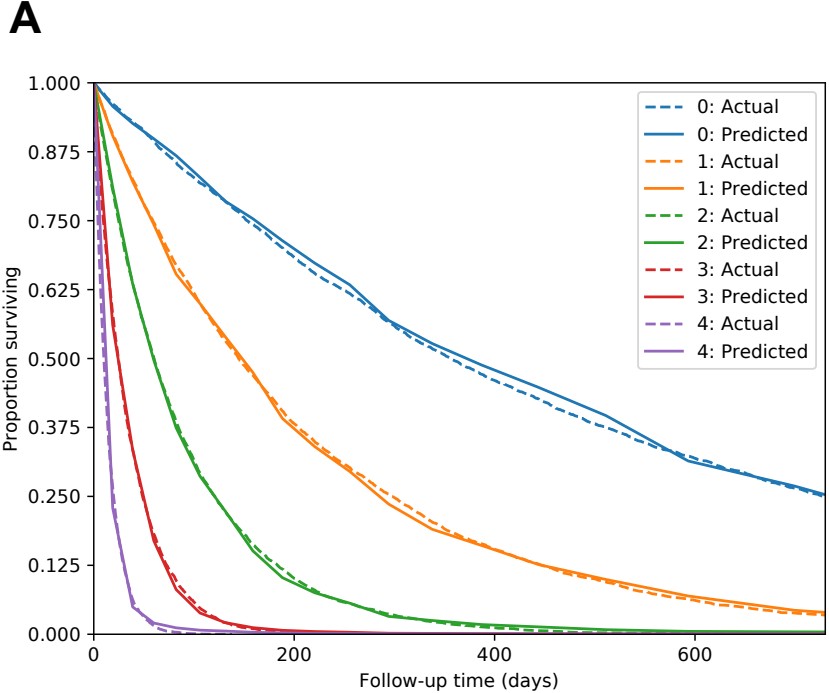

**B**

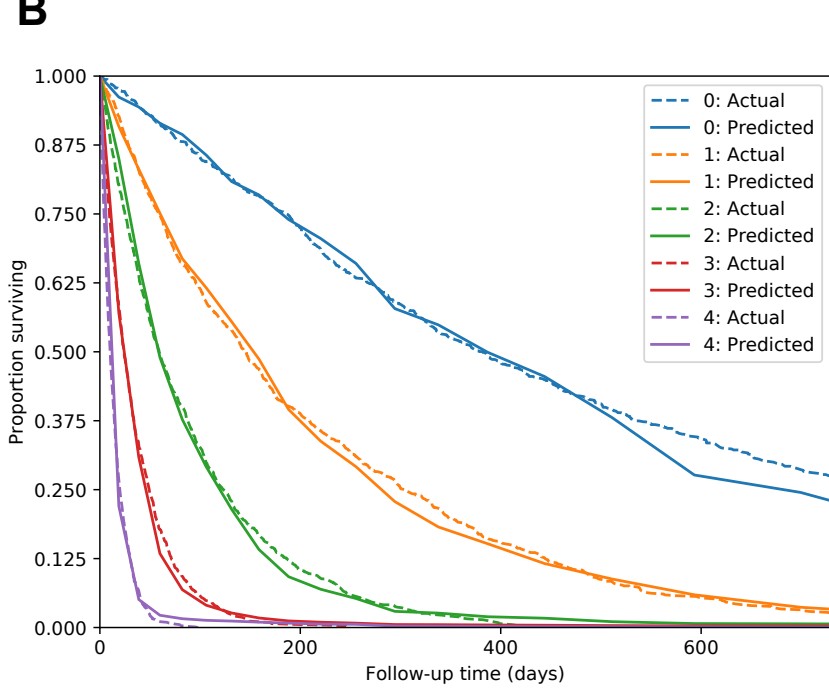

**Figure 4  MNIST dataset calibration plots for training set (A) and test set (B).** Images of handwritten digits 0–4 were used as predictor of survival time. Lower digits have longer median survival. For each digit, actual survival curve plotted with dotted line and mean model-predicted survival plotted with solid line.

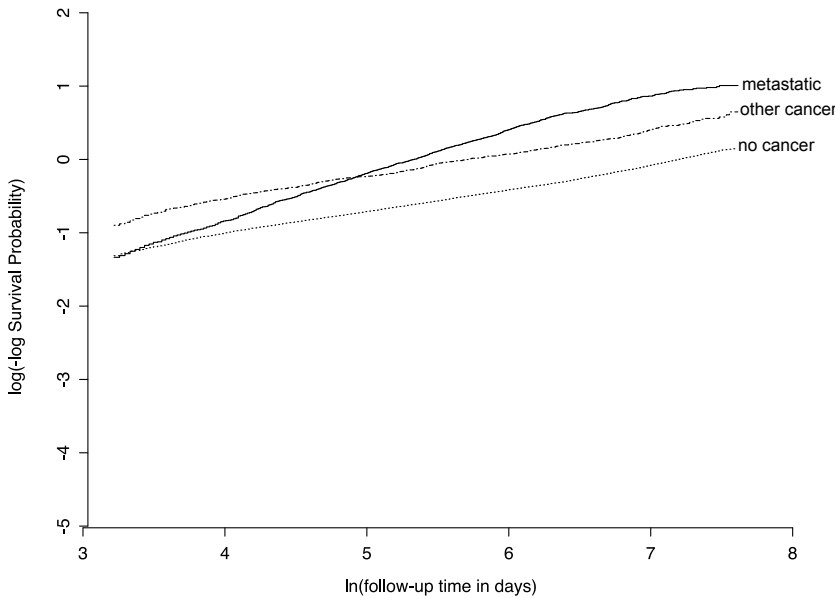

**Figure 5** **Example of violation of proportional hazards assumption in SUPPORT study dataset.** For the "ca" variable, patients with metastatic cancer have a similar risk of early death as other patients, but a higher risk of late death, as evidenced by non-parallel lines on this plot of log(-log survival) over time.

# DISCUSSION

We presented Nnet-survival, a discrete-time survival model for neural networks. It is theoretically justified since the likelihood function is used as the loss function, and naturally incorporates non-proportional hazards. Because it is a parametric model, it can be trained with mini-batch gradient descent as the likelihood/loss depends only on the patients in the current mini-batch. This enables fast training, use on datasets that do not fit in memory, and can avoid the network getting stuck in a local minimum of the loss function (*Bottou, 1991*). This is in contrast to models based on the Cox proportional hazards model such as Cox-nnet (*Ching, Zhu & Garmire, 2018*) and Deepsurv (*Katzman et al., 2018*), which require the entire training set to be used for each model update (batch gradient descent). The Nnet-survival model can be applied to a variety of neural network architectures, including multilayer perceptrons and convolutional neural networks.

In our experiments, the model performed well on both simulated and real datasets. It was challenging to find a publicly available dataset that would potentially highlight the advantages of the model. Ideally, such a dataset would have large sample size, predictor data such as images or text that are well-suited to neural networks, and time-to-event outcome data. Since no such dataset was available to our knowledge, we used the SUPPORT study dataset of 9,105 hospitalized patients, which has moderate sample size and time-to-event outcome data, but has low-dimensional predictor data that may not result in a benefit from a neural network approach. For this dataset, our model's discrimination and calibration

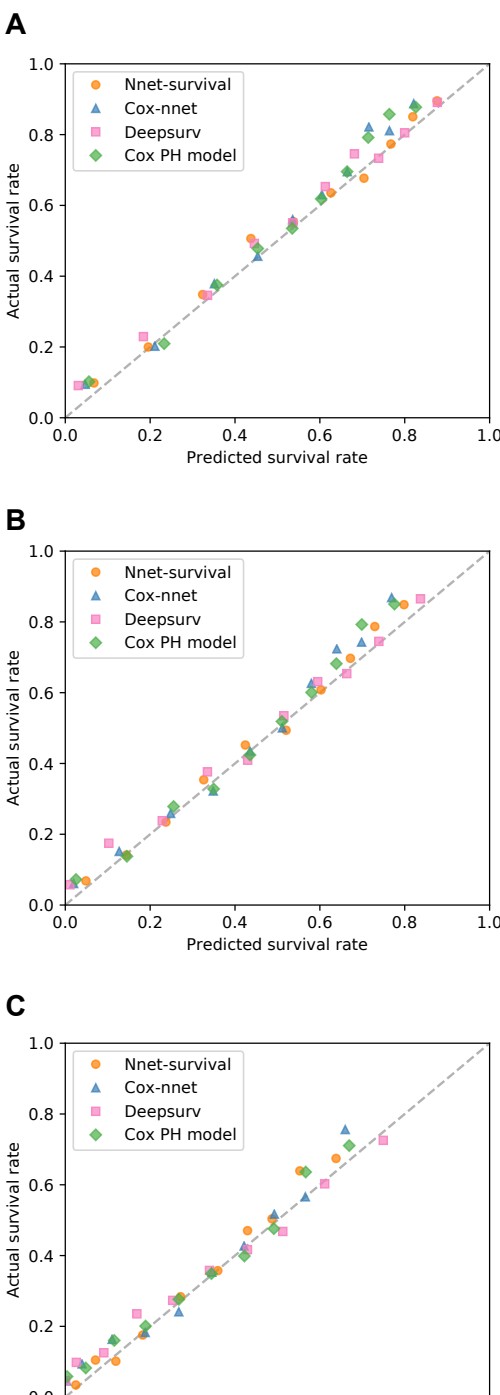

**Figure 6** **Calibration of four survival models on SUPPORT study test set.** For each model, for each of three follow-up times, patients were grouped by deciles of predicted survival probability. Then, for each decile, mean actual survival (Kaplan–Meier method) was plotted against mean predicted survival. A perfectly calibrated model would have all points on the identity line (dashed). Follow-up times: (A) 6 months; (B) 1 year; (C) 3 years.

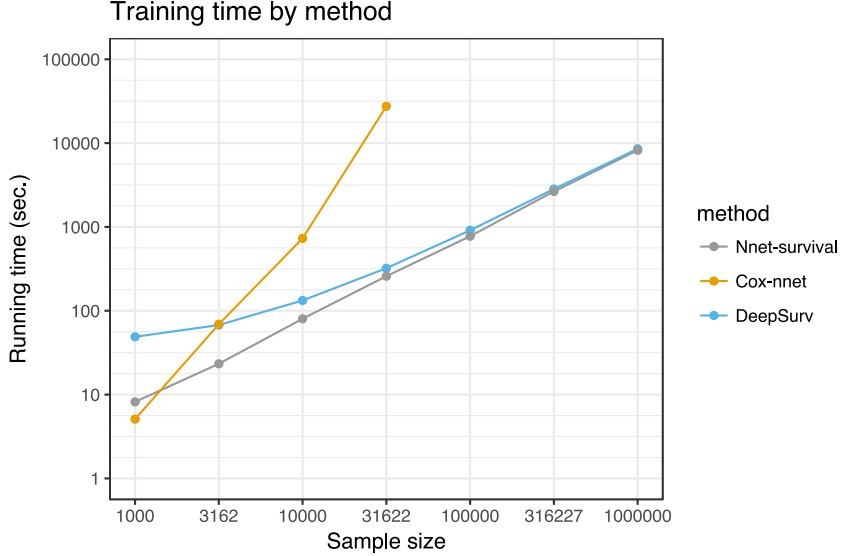

**Figure 7** **Running time of the three neural network models on SUPPORT study dataset.** Each point represents the average of three runs. Cox-nnet ran out of memory for sample sizes of 100,000 and higher.

performance was similar to several other neural network survival models and a traditional Cox proportional hazards model. In running time tests, its running time was similar to Deepsurv (*Katzman et al., 2018*) and better than Cox-nnet (*Ching, Zhu & Garmire, 2018*) for sample sizes >1,000. Interestingly, Cox-nnet ran out of memory for larger dataset sizes, because it stores an $n$ by $n$ matrix where $n$ is the sample size (variable name *R_matrix_train* in the Cox-nnet code).

While our model has several advantages and we think it will be useful for a broad range of applications, it does has some drawbacks. The discretization of follow-up time results in a less smooth predicted survival curve compared to a non-discrete parametric survival model such as a Weibull accelerated failure time model. As long as a sufficient number of time intervals is used, this is not a large practical concern—for instance, with 19 intervals the curves in Fig. 6 appear very smooth. Unlike a parametric survival model, the model does not provide survival predictions past the end of the last time interval, so it is recommended to extend the last interval past the last follow-up time of interest.

The advantages of parametric survival models and our discrete-time survival model could be combined in the future using a flexible parametric model, such as the cubic spline-based model of Royston and Parmar, implemented in the *flexsurv* R package (*Royston & Parmar, 2002*; *Jackson, 2016*). Complex non-proportional hazards models can be created in this way, and likely could be implemented in deep learning packages.

## CONCLUSIONS

Our discrete-time survival model allows for non-proportional hazards, can be used with stochastic gradient descent, allows rapid training time, and was found to produce good

discrimination and calibration performance with both simulated and real data. For these reasons, it may be useful to medical researchers.

### Funding

Balasubramanian Narasimhan's work supported in part by the Clinical and Translational Science Award 1UL1 RR025744 for the Stanford Center for Clinical and Translational Education and Research (Spectrum) from the National Center for Research Resource.

### Grant Disclosures

The following grant information was disclosed by the authors:
National Center for Research Resource: 1UL1 RR025744.

### Competing Interests

The authors declare there are no competing interests.

### Author Contributions

- Michael F. Gensheimer conceived and designed the experiments, performed the experiments, analyzed the data, prepared figures and/or tables, authored or reviewed drafts of the paper, approved the final draft.
- Balasubramanian Narasimhan conceived and designed the experiments, authored or reviewed drafts of the paper, approved the final draft.

### Data Availability

The code for this article is hosted at GitHub: https://github.com/MGensheimer/nnet-survival.

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
