# Peer review of "A scalable discrete-time survival model for neural networks"

_PeerJ, doi:10.7717/peerj.6257_

## Round 0.1 · original submission · Major Revisions

The two reviewers pointed out that both the structure of the manuscript and the experimental evaluation can be improved. One reviewer asked to make the dataset available. Please carefully address these major comments when preparing a revision.

Reviewer 1 ·

Basic reporting

The paper is written in professional English.
The literature references are properly summarized.
The paper follows a professional article structure.

Experimental design

The problem studied is within the aims and scope of PeerJ.
Researcher questions are meaningful and well-defined.
The experiments are not well designed. The description of section results-simulated data is very unclear. The authors should describe how data is simulated, and what the corresponding results are. Instead, the authors describe a mixture of ML models together with unclear results. I have checked the part for a couple of times, and I still don't understand it very well. Moreover, for the MNIST part of the simulated data, would the authors be more clear about this statement: "The ten digits were condensed into five classes, with shorter survival for higher numbers"?

Validity of the findings

no comment

Additional comments

Besides the unclear experimental design, I have two other major comments.
1) In the second paragraph of the Introduction, the authors motivate their work based on three points. To me, the first two motivations are not valid at all, i.e.,
- Few of the authors have published source code
- only some of the models are implemented in modern frameworks such as TensorFlow and Keras
I believe that these problems can be easily solved by some not-very-complicated engineering work.
2) In the description of the experimental results, the authors mention that "which was slightly inferior to the neural network survival model", this means the proposed neural network survival model is not really advancing the state-of-the-art. In this way, I don't see how this paper can meet the standard of a PeerJ publication.

Reviewer 2 ·

Basic reporting

This manuscript needs significant re-organization. A lot of wording should occur in the Methods section appears in the Results section. And some sections of Results read like manual/protocol, rather than research results (eg. line 230-246)

The Stanford Medical System metastatic cancer data do not appear to be public/shared.

Experimental design

As mentioned above, some data are not public to be replicable.

More comparisons with other neural network based methods, including deepSurv and cox-nnet, need to be demonstrated (for both accuracy and running time efficiency).

Validity of the findings

At various places, the statements need to be backed up by result demonstration.
For example, the authors stated that one must use fix time intervals (yet have similar numbers of events) vs. alternative approach where the number of events is fixed but the time interval length varies (line 186-189).

Additional comments

The reviewer appreciate the idea behind this manuscript. It is certainly appealing to supplement a NN based model when the data violates the proportional hazards assumption in the cox-PH model. It is also neat to have trained minibatch in each discrete time to enable fast training of the models. The authors did a good job at motivating this approach. However, there are various technical and organizational issues of this manuscript.

1. The data sets used in this manuscript are yet to demonstrate that the Cox-PH model's assumptions fail on them, suggest to provide the evidence in supplementary figures.

2. There is no comparison with other neural network/ deep learning based Cox-PH extension methods, including DeepSurv and Cox-nnet. The authors need to compare their method with these two methods, in terms of accuracy and speed.

3. The authors described 4 simulated data sets, however only show results on 1 data set, they need to show all 4 data (otherwise why mentioning them?)

4. Various places, the authors made claims without showing the evidence, please include evidence as supplementary figures. For example, what supports the statement on line 186-189?

5. How many time intervals exactly to have in order to have the best performing method? Rather than a range of 30-50? Any ways to optimize it for the reader, rather than being a user supplied number?

6. This manuscript needs significant re-organization. A lot of wording should occur in the Methods section appears in the Results section. And some sections of Results read like manual/protocol, rather than research results (eg. line 230-246).

7. The Stanford Medical System metastatic cancer data do not appear to be public/shared. This violates peerJ policy.

---

## Round 0.2 · accepted · Accept

Although neither prior reviewer was able to review this submission, I confirm that the revisions are acceptable. The manuscript is ready for publication.